# Exposure to Non-Steady-State Oxygen Is Reflected in Changes to Arterial Blood Gas Values, Prefrontal Cortical Activity, and Systemic Cytokine Levels

**DOI:** 10.3390/ijms25063279

**Published:** 2024-03-14

**Authors:** Elizabeth G. Damato, Joseph S. Piktel, Seunghee P. Margevicius, Seth J. Fillioe, Lily K. Norton, Alireza Abdollahifar, Kingman P. Strohl, David S. Burch, Michael J. Decker

**Affiliations:** 1Department of Physiology & Biophysics, Center for Aerospace Physiology, School of Medicine, Case Western Reserve University, Cleveland, OH 44106, USA; egd@case.edu (E.G.D.);; 2Department of Emergency Medicine, MetroHealth Medical Center, School of Medicine, Case Western Reserve University, Cleveland, OH 44106, USA; 3Department of Population and Quantitative Health Sciences, School of Medicine, Case Western Reserve University, Cleveland, OH 44106, USA; 4Cuyahoga Community College, Cleveland, OH 44122, USA; 5Jacobs School of Medicine and Biomedical Sciences, University at Buffalo, Buffalo, NY 14203, USA; 6711th Human Performance Wing, Biomedical Impact of Flight Branch, Air Force Research Laboratory, United States Air Force, Wright-Patterson Air Force Base, Dayton, OH 45433, USA

**Keywords:** hypobaric, oxygen, arterial, brain, neurovascular, spectroscopy

## Abstract

Onboard oxygen-generating systems (OBOGSs) provide increased inspired oxygen (F_i_O_2_) to mitigate the risk of neurologic injury in high altitude aviators. OBOGSs can deliver highly variable oxygen concentrations oscillating around a predetermined F_i_O_2_ set point, even when the aircraft cabin altitude is relatively stable. Steady-state exposure to 100% F_i_O_2_ evokes neurovascular vasoconstriction, diminished cerebral perfusion, and altered electroencephalographic activity. Whether non-steady-state F_i_O_2_ exposure leads to similar outcomes is unknown. This study characterized the physiologic responses to steady-state and non-steady-state F_i_O_2_ during normobaric and hypobaric environmental pressures emulating cockpit pressures within tactical aircraft. The participants received an indwelling radial arterial catheter while exposed to steady-state or non-steady-state F_i_O_2_ levels oscillating ± 15% of prescribed set points in a hypobaric chamber. Steady-state exposure to 21% F_i_O_2_ during normobaria produced arterial blood gas values within the anticipated ranges. Exposure to non-steady-state F_i_O_2_ led to P_a_O_2_ levels higher upon cessation of non-steady-state F_i_O_2_ than when measured during steady-state exposure. This pattern was consistent across all F_i_O_2_ ranges, at each barometric condition. Prefrontal cortical activation during cognitive testing was lower following exposure to non-steady-state F_i_O_2_ >50% and <100% during both normobaria and hypobaria of 494 mmHg. The serum analyte levels (IL-6, IP-10, MCP-1, MDC, IL-15, and VEGF-D) increased 48 h following the exposures. We found non-steady-state F_i_O_2_ levels >50% reduced prefrontal cortical brain activation during the cognitive challenge, consistent with an evoked pattern of neurovascular constriction and dilation.

## 1. Introduction

Hypobaric hypoxia and decompression injury are among the risks awaiting tactical aviators within their austere, high-altitude environment. The life-support equipment providing a continuous fraction of inspired oxygen (F_i_O_2_), typically between 35 and 100%, is an intended neuroprotection against those hazards. Exposure to enhanced oxygen concentrations also conveys a risk. Prolonged exposure is pathologic to multiple organ systems [1,2] while short term exposure reduces cerebral perfusion and alters cortical electroencephalographic activity [3].

The original technology to provide the increased F_i_O_2_ levels necessary for survival within high altitude environments was first implemented during World War I. Those systems used liquid oxygen, a relatively simple approach [4,5]. As the liquid oxygen transitioned into a gaseous state, it flowed into the aircrew’s breathing system. There, it was diluted with air to deliver a specific level of F_i_O_2_ to the aviator, with that level determined by cockpit ambient pressure [6]. Further dilution of the F_i_O_2_ could occur within the gas regulators, such as that found within diluter demand masks. The success of liquid-oxygen systems led to their further development and, beginning in the early 1940s, the installation of a small refillable liquid oxygen container within almost every tactical aircraft. The simplicity of the liquid-oxygen systems contributes to their continued use.

Onboard oxygen-generation systems (OBOGs) began replacing liquid-oxygen-based life-support systems during the 1970s [7]. Their operation is dependent upon a stream of ambient air being forced through a molecular sieve. Only oxygen molecules pass through the sieve and into a series of small storage tanks [8]. Upon release from the storage tanks, dilution occurs in diluter-demand regulators. The criteria of both minimum and maximum F_i_O_2_ delivery are determined according to the aircraft cockpit pressure. The OBOGs typically incorporate “safety pressure”, which is a continuous positive pressure within the pilot’s mask of 1.25–4 cm of water (cmH_2_O). Systems without safety pressure, or when safety pressure is turned off, will provide 0–0.5 cmH_2_O pressure within the mask. In contrast with liquid-oxygen systems that deliver a relatively constant F_i_O_2_ concentration into the life support system, the F_i_O_2_ delivery from the OBOGs system follows a sinusoidal pattern [6].

Oxygen-induced reductions in cerebral perfusion, and subsequently, oxygen delivery to the brain, was first characterized in healthy humans with indwelling arterial and venous catheters by Lambertsen et al., in 1953 [9]. Those findings have since been replicated using noninvasive techniques [10,11], with more recent studies revealing that reductions in cerebral perfusion begin during exposure to an F_i_O_2_ of 60% [12]. Inspired oxygen levels > 60% not only influence neurovascular tone; systemic sequelae can range from decreased cardiac output [13] and the onset of lung alveolar collapse and atelectasis [14], to tonic-clonic seizure activity when hyperoxia occurs during hyperbaric conditions [15].

In contrast with well characterized neuro- and physiological outcomes of exposure to steady-state F_i_O_2_, fewer studies have focused upon the outcomes of exposure to non-steady-state F_i_O_2_. Those findings, often derived from cell [16], rodent, or swine models [17] reveal inflammation of the pulmonary epithelium. Those observations concur with studies by Formenti et al. [18,19] revealing that non-steady-state F_i_O_2_ levels, induced by altering the ventilatory inspiratory and expiratory ratios, lead to non-steady-state arterial blood oxygen (P_a_O_2_) levels [17]. While those studies affirm that exposure to non-steady-state F_i_O_2_ induces non-steady-state P_a_O_2_ levels, there is an absence of peer-reviewed publications assessing the impact of non-steady-state F_i_O_2_ and P_a_O_2_ upon functional activity within the human cerebral cortex. Our objective for this study was to address that void.

To inform our understanding of the central nervous system’s functional response to non-steady-state F_i_O_2_, we measured prefrontal cortical activity during a cognitive task immediately following exposure to non-steady F_i_O_2_ and again 120 s later following exposure to steady-state F_i_O_2_. This enabled us to test the following hypotheses: (1) Following exposure to non-steady-state F_i_O_2_, arterial blood oxygen (P_a_O_2_) levels will reflect the F_i_O_2_ at the time which the sample was obtained; (2) The prefrontal cortex will achieve a different level of activation during a cognitive task following exposure to non-steady-state F_i_O_2_ when compared to exposure to steady-state F_i_O_2_; and (3) Exposure to increased F_i_O_2_ levels, both steady and non-steady-state, will be followed by increased levels of systemic proinflammatory serum analytes.

## 2. Results

Thirty-seven candidates received a verbal description of the study protocol and provided written informed consent to participate. Of those 37, seven withdrew prior to participating due to scheduling conflicts or concerns regarding COVID-19. Of the remaining 30 people who began the study, six were excluded from continuing the protocol following a vasovagal response during arterial catheter placement. The remaining 24 participants completed all phases of the protocol. Their basic demographics are presented in Table 1; there were no differences between male and female study participant’s age or BMI.

### 2.1. Non-Steady-State/Steady-State F_i_O_2_ Exposure

Continuous tracing from an oxygen sensor attached to the study participant’s facemask validated that the F_i_O_2_ delivered to the participant matched the intended output from the air/oxygen blender. The signal hysteresis depicted in Figure 1, especially at the top of each square wave, represents nitrogen washout from the lungs, which can take several minutes [20]. The hysteresis/nitrogen washout reveals the heterogeneity in lung ventilation and confirms that a steady-state, homogenous level of oxygen within the lungs was not achieved during the 60 s cycle time of each F_i_O_2_ oscillation.

This graph displays the output from the mask oxygen sensor from one study participant. The *x*-axis is time in minutes with each block representing two minutes. The *y*-axis is oxygen concentration measured within the mask, with each block representing 8.0 percent. The blue circles represent the time points of each arterial blood sample. The series of three pegboard tests (not indicated on this figure) were initiated after each collection of each arterial blood sample.

### 2.2. Arterial Blood Gas Results

Table 2 illustrates that during normobaric ambient pressure (749 mmHg), the P_a_O_2_ levels measured during exposure to steady-state 21% F_i_O_2_ were 95.63 ± 13.21 mmHg (mean ± 1 SD), while exposure to steady-state 100% F_i_O_2_ produced P_a_O_2_ levels of 541.83 ± 47.62 mmHg. Those values were within the anticipated range for healthy people [21]. Table 2 also presents the P_a_O_2_ levels measured during exposure to steady-state 21% and 100% F_i_O_2_ at the simulated altitudes of 8000 ft (565 mmHg) and 15,000 ft (494 mmHg). Those P_a_O_2_ levels were also within the anticipated ranges [22]. The accompanying values of P_a_CO_2_, pH, HCO_3_, S_a_O_2_, glucose, and hematocrit are provided in Appendix A.

The values of P_a_O_2_ measured at the conclusion of each non-steady-state F_i_O_2_ sequence were significantly greater than the P_a_O_2_ values measured 120 s later during exposure to steady-state F_i_O_2_. This pattern was consistent during all non-steady-state F_i_O_2_ sequences administered at 749 mmHg (normobaric conditions), 565 mmHg (8000 ft), and 494 mmHg (15,000 ft). However, the magnitude of difference between the P_a_O_2_ values observed immediately post non-steady-state F_i_O_2_ with those observed following 120 s of steady-state F_i_O_2_ increased as ambient environmental pressure decreased. Whereas the mean percent change was 12.81% during normobaric conditions at 749 mmHg, that mean percent change increased to 15.37% for P_a_O_2_ values measured at 565 mmHg and 17.55% for P_a_O_2_ values measured at 494 mmHg.

### 2.3. fNIRS Results

Table 3 reports comparisons of fNIRS contrasts performed during the grooved pegboard task immediately following exposure to oscillatory F_i_O_2_ with those performed during a grooved pegboard task following 120 s of exposure to steady-state F_i_O_2_. We found exposure to non-steady-state cyclic F_i_O_2_ > 50% and < 100% led to reduced prefrontal cortical activation levels immediately following 5 min of exposure to oscillatory F_i_O_2_ at both normobaric pressure and during exposure to a reduced barometric pressure of 494 mmHg. These results are graphically illustrated in Figure 2.

#### 2.3.1. Pressure of 749mmHg (Normobaria)

No differences in levels of prefrontal cortical activation occurred at normobaric pressure between Task #1 and Task #3 during exposure to a steady-state F_i_O_2_ of 21%. The prefrontal cortical activation levels also did not differ between pegboard Task #1 and Task #3 during exposure to a non-steady-state F_i_O_2_ of 35% ± 15% and F_i_O_2_ of 50% ± 15%. However, significant differences in prefrontal cortical activation did emerge between Task #1 and Task #3 during exposure to a non-steady-state F_i_O_2_ of 65% ± 15% as well as F_i_O_2_ of 80% ± 15%. At those oxygen concentrations, prefrontal cortical activation levels were significantly lower immediately after exposure to non-steady-state F_i_O_2_ (Task #1) than during steady-state F_i_O_2_ (Task #3). Those differences resolved upon exposure to a steady-state F_i_O_2_ of 100%.

#### 2.3.2. Pressure of 565 mmHg (8000 ft)

Prefrontal cortical activation levels only differed between Task #1 and Task #3 at 565 mmHg following exposure to a steady-state F_i_O_2_ of 21%. Exposure to increased oxygen concentrations > 21%, presented as either non-steady-state or steady-state, did not elicit differences in prefrontal cortex activation levels between pegboard Task #1 and Task #3.

#### 2.3.3. Pressure of 494 mmHg (15,000 ft)

Prefrontal cortical activation levels at 494 mmHg differed between Task #1 and Task #3 during exposure to a steady-state F_i_O_2_ of 21%. Prefrontal cortical activation levels also differed between Task #1 and Task #3 while the participant was exposed to non-steady-state F_i_O_2_ levels of 50% ± 15% and F_i_O_2_ levels of 80% ± 15%. Those differences resolved upon exposure to steady-state F_i_O_2_ of 100%.

### 2.4. Blood Serum Results

The levels of 37 serum analytes were measured in blood samples obtained at each of the two data collection time points. Of those 37 analytes, the serum levels of six analytes differed significantly between the values obtained prior to exposure to increased F_i_O_2_ and hypobaria with those values measured 48 h after completion of all exposures. Serum levels for those six analytes and the outcomes of the statistical comparisons are provided in Table 4. This table reveals that IL-6, measured by the proinflammatory plate, was increased. The chemokine plate showed increases in IP-10, MCP-1, and MDC. The cytokine plate revealed increased levels of IL-15 while the angiogenesis plate demonstrated that VEGF-D was also increased post exposure.

## 3. Discussion

Our objective was to characterize physiologic responses to exposure to both non-steady-state and steady-state F_i_O_2_ during normobaric and hypobaric environmental pressures. The primary outcomes included serial measurements of arterial blood gases (ABG) and prefrontal cortical activity during cognitive challenge. The secondary outcomes included measures of general blood chemistry and the quantification of selected serum analytes within the circulatory system both before and after exposure.

The results indicate that steady-state exposure to 21% F_i_O_2_ during normobaric environmental pressures produced ABG values within the anticipated ranges (control condition). Exposure to non-steady-state F_i_O_2_ levels led to transient elevations of P_a_O_2_ values that were higher upon cessation of non-steady-state F_i_O_2_ than when measured 120 s later during steady-state exposure to the same F_i_O_2_ level (Table 2). This pattern of increased P_a_O_2_ levels following exposure to non-steady-state F_i_O_2_ was consistent across all F_i_O_2_ ranges, and present at each of the three barometric pressure conditions.

When exposed to both steady-state and non-steady-state F_i_O_2_ levels < 50%, prefrontal cortical activation measured during pegboard Task #1 was not different from levels measured 120 s later during pegboard Task #3. In contrast, exposure to non-steady-state cyclic F_i_O_2_ > 50% and <100% led to reduced prefrontal cortical activation levels during Task #1 when compared with levels measured 120 s later during Task #3. These differences emerged at both normobaric pressure and during exposure to a reduced barometric pressure of 494 mmHg, which simulated an altitude of 15,000 feet.

In addition to acute transient changes in the ABG values and levels of prefrontal cortical activation, we also observed that six serum analyte levels were increased 48 h following exposure to increased F_i_O_2_ and hypobaria. Those included IL-6, IP-10, MCP-1, MDC, and IL-15. The vascular endothelial growth factor, VEGF-D, was also increased.

Arterial blood gas values of P_a_O_2_, measured within two to three breaths following cessation of non-steady-state F_i_O_2_, suggest that the oxygen content within each inspired breath was rapidly conveyed into the circulatory system. During exposure to non-steady-state F_i_O_2_ levels of 65% ± 15% F_i_O_2_, an atmospheric pressure of 749 mmHg, a P_a_CO_2_ of 41.9 mmHg with presumed partial pressure of water vapor in the airway of 47 mmHg, and a respiratory quotient (R.Q.) of 0.8, the alveolar pressure of oxygen (PAO_2_) was predicted to be 403.9 mmHg. When accounting for the alveolar–arterial (Aa) gradient, which reduces the partial pressure of oxygen by 5–10 mmHg, the anticipated P_a_O_2_ within those participants should have been between 389.9 and 393.9 mmHg. Table 2 reveals that following the non-steady-state F_i_O_2_ sequence of 65% ± 15%, the actual P_a_O_2_ values were 382.5 ± 42.9 mmHg (mean ± 1 SD), which closely approximates to the expected value.

In a recently reported study [23], human participants were exposed to 60 s oscillations of 80% F_i_O_2_ and 20% nitrogen (N_2_) and between 70% F_i_O_2_ and 30% N_2_ in a hypobaric chamber at 8000 feet in two exposure cycles of 45 min alternating with a 45 min break. Measurements were taken at ground level before, between, and after the oscillatory exposures and comparisons were made between baseline and hyperoxic oscillatory exposures while remaining at 8000 ft. No differences were found in arterial P_a_O_2_ across the measurement time points but statistically significant reductions in P_a_CO_2_ were noted between and following the oscillatory exposures. Conversely, our study compared P_a_O_2_ and P_a_CO_2_ levels following 120 s exposures of steady-state F_i_O_2_ of 65% and 80% to P_a_O_2_ and P_a_CO_2_ levels obtained following three 60 s oscillations ± 15% of those respective steady-state values at 8000 feet. Key differences between the two experimental designs prevent meaningful comparisons of our study with the Kelley et al. publication. Additional barriers to comparing the two studies include the lack of explanation for the very low P_a_CO_2_ levels at baseline and the omission of pH, bicarbonate, and other ABG components that would allow interpretation. Ambiguity as to whether baseline blood gas measurements used for statistical comparisons were obtained at sea level or at 8000 feet prior to oscillatory exposure also limit inter-study comparisons.

If significant mixing occurred between the newly inspired oxygen content contained within each inspiration with the pre-existing oxygen content within the residual lung volume, the P_a_O_2_ value measured following each non-steady-state F_i_O_2_ sequence would have been much higher. During the non-steady-state F_i_O_2_ sequence of 65% ± 15%, the alveolar air equation estimates that an F_i_O_2_ of 80%, which was the peak F_i_O_2_ prior to the drop to 65% F_i_O_2_, would produce P_a_O_2_ values of 499–504 mmHg [24]. If “mixing” occurred between an existing residual lung volume of gas containing 80% F_i_O_2_ with an incoming gas containing an F_i_O_2_ of 65%, the predicted P_a_O_2_ would have been ~440.5 mmHg. However, actual P_a_O_2_ levels were not in that range; they were instead 382.5 ± 42.9 mmHg. This suggests that during the two to three breaths occurring during the transition from 80% F_i_O_2_ to 65% F_i_O_2_, the newly inspired oxygen content was immediately transferred from the alveoli into the circulatory system. This would be consistent with observations by Formenti and Farmery [19], who revealed breath-by-breath oscillations in P_a_O_2_ do occur, and those oscillations correspond with the inspiratory and expiratory phases of the respiratory cycle. Their studies suggest that the actual oxygen content within each inspired breath is rapidly transferred across the alveolar epithelium and into the systemic circulation.

Exposure to non-steady-state F_i_O_2_ sequences evoked a systematic and reproducible outcome on P_a_O_2_ levels, regardless of ambient environmental pressures. In contrast, non-steady-state F_i_O_2_ influences upon prefrontal cortical activation were less systematic and were constrained to F_i_O_2_ exposures >50% but <100%. The potential mechanisms influencing these changes in prefrontal cortical activation levels have been informed by studies characterizing the relationship between increased F_i_O_2_ with cerebral perfusion. Those studies [3,12], conducted under normobaric conditions, revealed that cerebral perfusion is unaffected by F_i_O_2_ levels ≥ 21% but ≤50%. Beginning with F_i_O_2_ levels of 60%, cerebral perfusion begins to decline and continues to fall with each incremental increase in F_i_O_2_. Upon reaching an F_i_O_2_ of 80%, cerebral perfusion is reduced by ~25% of baseline values (mean ± 1 SD), dropping from 46.16 ± 10.11 milliliters per minute per 100 g of tissue (mL/min/100 g) to 34.58 ± 8.59 mL/min/100 g. During exposure to 100% F_i_O_2_, cerebral perfusion is 32.46 ± 7.24 mL/min/100 g, a five percent reduction from levels observed during exposure to 80% F_i_O_2_. Collectively, these findings suggest that exposure to F_i_O_2_ levels of ~60% evokes onset of neurovascular constriction. Increasing F_i_O_2_ beyond 60% prompts further neurovascular constriction, which becomes maximal during exposure to F_i_O_2_ levels between 80% and 100%.

Our collective observations from this and prior studies suggest that (1) P_a_O_2_ levels are increased following exposure to non-steady-state F_i_O_2_ compared to when measured following the subsequent 120 s of steady-state F_i_O_2_ (Table 2), and (2) F_i_O_2_ levels ≥ 60% reduce cerebral perfusion via neurovascular constriction [12], and finally, (3) lowering the F_i_O_2_ from 100% to 21% leads to neurovascular dilation with the restoration of cerebral perfusion to baseline levels [12]. Those findings lead us to suspect that exposure to non-steady-state, cyclic F_i_O_2_ levels within the range of 65% ± 15% induces non-steady-state, cyclic neurovascular constriction and dilation. During exposure to 80% F_i_O_2_, the peak level delivered during the non-steady-state F_i_O_2_ sequence of 65% ± 15%, maximal neurovascular constriction with a concomitant reduction in cerebral perfusion would have existed. We believe the reduced levels of prefrontal cortical activity observed during pegboard Task #1 reflected the reduced cerebral perfusion occurring at the peak F_i_O_2_ during non-steady-state exposures.

Due to the inherent circulatory delay between the lungs and brain [25,26], the change in F_i_O_2_ from 80% to 65% followed by neurovascular dilation with increased cerebral perfusion, would not have occurred until after completion of pegboard Task #1. Beginning with pegboard Task #3, which commenced following 120 s of steady-state F_i_O_2_, neurovascular dilation with increased cerebral perfusion would have achieved a constant level. The relative increase in cerebral perfusion at F_i_O_2_ of 65% versus that following exposure to an F_i_O_2_ of 80% would have led to increased oxygen delivery. We believe that during pegboard Task #3, the increase in cerebral perfusion and oxygen delivery could have accounted for the increased prefrontal cortical activation levels that we observed. While that hypothesis is biologically plausible, confirming or refuting it would have required additional studies and experimental techniques on cortical connectivity that were beyond the intent and scope of this study.

In addition to the changes in P_a_O_2_ and levels of prefrontal cortical activation that emerged during exposure to non-steady-state F_i_O_2_, we also observed increased levels of six blood serum analytes 48 h later, following the conclusion of the experimental protocol. Comparison of our serum analyte levels to previously published reports is limited by different analysis techniques (enzyme-linked immunosorbent assay (ELISA)) vs. the more sensitive multi-array technology we used) and the scarcity of studies documenting inter-subject variation and temporal changes in baseline levels within non-clinical populations. Two studies conducted in healthy individuals concluded that significant variability exists in the baseline levels between healthy individuals, and most cytokines remain stable across serial measurements [27,28]. The baseline and post-exposure serum analyte levels we report appear to reside within published normative reference range measurements [27,29,30,31,32]. However, comparisons are limited between ELISA-derived values and those obtained from ultrasensitive assay methods that are essential for characterizing cytokine levels [28]. More importantly, the increase in the serum analyte levels in our study concurs with prior observations that exposure either to steady state or non-steady-state F_i_O_2_ levels ≥ 50% elicits the onset of proinflammatory biochemical cascades within the pulmonary epithelium and other cell types [17]. In addition, the placement of the radial artery catheter could also have initiated the release of several of the proinflammatory serum analytes presented in Table 4. Although baseline levels of MCP-1 have been reported to vary across time in healthy individuals [27], MCP-1 influences the movement of monocytes out of the bloodstream, across the endothelium and into the tissues to engage in immunologic surveillance and inflammatory responses [33] and could explain the increase we observed. Determining the specific mechanisms underlying increased analyte levels is beyond the scope of this project but may represent an adaptive or maladaptive physiological stress response.

### Strengths and Limitations

The intent of this study was to characterize the physiologic responses of healthy individuals to exposure to both steady-state and non-steady F_i_O_2_ levels above 21%. Specific F_i_O_2_ levels were delivered in a non-steady-state, square wave cycle which ranged ±15% around a pre-specified level or “dose” of oxygen. That experimental design was intended to model oxygen delivery provided by aviation life support systems that employ onboard oxygen-generating systems, some of which can potentially deliver oxygen concentrations of ±15% around a pre-defined level. Study participants were exposed to those non-steady F_i_O_2_ levels within a hypobaric chamber that provided three different barometric pressures: 749 mmHg, which was the normobaric pressure of the study location, 565 mmHg paralleling an altitude of 8000 feet, and 494 mmHg, which was equivalent to an altitude of 15,000 feet. Each pressure was chosen as representative of cockpit pressures encountered by tactical aviators. The simultaneous and conscious manipulation of the key experimental variables of F_i_O_2_ and environmental pressure could induce inherent confounds to the data analyses and interpretation, and thereby, present a study weakness. In contrast, our a priori intent was to recreate an exposure matrix of non-steady-state F_i_O_2_ levels delivered within a normobaric and hypobaric environment, thereby emulating the life support system and ambient cockpit pressures that many tactical aviators routinely experience.

Statistical analyses revealed that P_a_O_2_ levels differed immediately following exposure to non-steady-state F_i_O_2_ compared to P_a_O_2_ levels 120 s later during steady-state exposure to steady-state F_i_O_2_. Levels of prefrontal cortical activity, which were measured at the same time as the ABG samples were taken, also differed but only during F_i_O_2_ exposures ≥ 60%. In addition, six of thirty-seven blood serum analytes differed between baseline and the study endpoint. However, our experimental design, which included placement of an arterial line catheter, precludes a defensible discussion of how hyperoxia/hypobaria may have contributed to the changes in the serum analytes that we observed.

It remains unclear why non-steady-state F_i_O_2_ levels > 50% but <100% led to changes in prefrontal cortical activation between Task #1 and Task #3 only at normobaric pressure (749 mmHg) and again at a barometric pressure of 494 mmHg, equivalent to 15,000 feet. Figure 2 suggests that at an atmospheric pressure of 565 mmHg simulating an altitude of 8000 feet, the pegboard Task #1 to Task #3 patterns of prefrontal cortical activation were similar to those seen during at an atmospheric pressure of 494 mmHg, equivalent to 15,000 feet. The difference was that the individual range of the prefrontal cortical activation levels at 565 mmHg between pegboard Task #1 and Task #3 was much greater than that observed at either 749 mmHg or 494 mmHg (Table 3). As the participants in our study had never before been inside a hypobaric chamber, their novel experience with depressurization may have led to a temporary state of heightened arousal that subsequently masked statistically changes in the prefrontal cortical activation between pegboard Task #1 and Task #3. Acclimatization to the experience of hypobaria at 565 mmHg could have reduced the likelihood of anxiety during the transition to the lower barometric pressure of 494 mmHg, enabling changes in the prefrontal cortical activation between pegboard Task #1 and Task #3 to be observed during that phase of the study. However, this is only speculation, and no defensible reason exists to exclude those persons who experienced the wider range of variability of prefrontal cortical activation during pegboard testing at 565 mmHg. While a sham condition may have reduced any confounding effect of participant anxiety, the invasiveness of arterial line placement and continued data collection during the decreasing but ever-present exposure risks of the pandemic, precluded this option.

To inform our decision as to whether Type I error was necessary, we identified recent studies of a similar sample size, N = 20 [34] and a larger N = 51 [35], using similar technologies to those employed in this study. Neither study applied Bonferroni corrections. Nonetheless, we felt that controlling for Type I error could enhance the rigor of our analyses and the defensibility of findings. As the application of Bonferroni corrections in a study such as this could obscure pertinent findings [36], we chose to present all significant findings and annotate those that remained significant following false discovery rate (FDR) correction. We consider this level of transparency to be a strength rather than a limitation of this study.

We do not believe that this study’s limitations impact the significance of our findings that exposure to non-steady F_i_O_2_ levels are followed by changes in arterial blood oxygen content, changes in prefrontal cortical activation levels, and enhanced systemic levels of proinflammatory serum analytes. Although neurovascular tone was not directly measured in this study, our findings pose the hypothesis that neurovascular constriction and dilation may potentially exist during exposure to non-steady-state F_i_O_2_ levels ≥ 60%. If it does indeed occur, executive functioning, cognitive performance, and visuomotor speed and accuracy could be impacted. Addressing that relevant question will require additional studies employing experimental protocols, cognitive assessments, and other physiologic measurement techniques beyond those employed in this study.

## 4. Materials and Methods

The objective of this study was to define the neurophysiologic and systemic responses to non-steady-state levels of inspired oxygen concentrations at barometric pressures of 749 mmHg (normobaric pressure at the study site), 565 mmHg (equivalent to an altitude of 2438 m or 8000 feet), and 494 mmHg (equivalent to an altitude of 4572 m or 15,000 feet). Measurements of ABG levels and prefrontal cortical brain activity were obtained from healthy volunteers during exposure to non-steady-state and steady-state F_i_O_2_.

The study protocol was approved by the Institutional Review Board of University Hospitals, STUDY20191540 and the United States Air Force Human Research Protection Office (HRPO), Protocol Number: FWR20200056X. Candidate study participants were required to be non-smokers between the ages of 18 and 61 years, and without exposure to a high-altitude environment >8000 ft. within the previous two weeks. Exclusion criteria included a medically conferred diagnosis of pulmonary and/or cardiac disease, a current or previous neurologic issue (e.g., epilepsy or seizure disorder), known history of sickle cell disease or sickle cell trait, elevated risk for bleeding or a bleeding disorder, recent (within the past 5 years) history of inner ear problems, a positive urine pregnancy test or currently attempting pregnancy, or a history of claustrophobia. Upon verifying the absence of exclusion criteria, all candidate participants were provided with, and then signed, an informed consent document.

To establish an appropriate sample size, we first conducted a series of preliminary studies in which patterns of prefrontal cortical activity were assessed with functional near infrared spectroscopy during a task-on and task-off cognitive challenge. During that testing, participants were exposed to both non-steady-state and steady-state F_i_O_2_ conditions, as described below.

Outcomes from those studies revealed that an N = 24 would provide sufficient data to achieve a minimum statistical power of 0.80 at a two-tailed significance of <0.05 to detect differences in prefrontal cortical activity during pegboard testing between exposures to non-steady-state and steady-state F_i_O_2_.

A detailed explanation of the experimental protocol and study methods follows (see Figure 3).

### 4.1. Health Screening and Protocol Familiarization

Upon arrival for the day of experimental exposure, participants were queried about current medications followed by measurements of height and weight, body temperature, vital signs, and self-reported outcome of their most recent coronavirus disease (COVID-19) test. Those assessments were followed by a focused neurological and physical examination performed by a licensed clinician.

Participants were next instructed on procedures that would occur during the experimental exposures. This included a practice session on the grooved pegboard test (Model 32025 Lafayette Instrument, Lafayette, IN, USA) which was employed to activate the prefrontal cortex using a task-on/task-off paradigm [37] executed in sequence three times. The pegboard contains 25 holes with randomly positioned slots in which metal pegs can be placed. The pegs have a key shape along one side and must be rotated to match the hole before they can be inserted. During the 20-s “task-on” phase, participants were instructed to pick up the pegs, one at a time, and place them into the board. When cued by the researcher for the 20-s “task-off” phase, participants were instructed to stop placing pegs, not move their hands or speak, and try to clear their minds. Participants were coached to not achieve a personal best during any of the tests. Rather, the goal was to complete the task as consistently as possible.

### 4.2. Indwelling Radial Arterial Catheter Placement

With the study participant seated comfortably, an Allen’s test was performed to ensure adequate collateral perfusion throughout both the right and left hands [38]. A small heating pad was then placed into the participant’s non-dominant hand to promote vasodilation of the target artery and the wrist was prepped and draped to provide a sterile field. A 1.0 milliliter (mL) subdermal injection of 1% lidocaine was injected into ventral aspect of the wrist to provide local anesthesia. Using ultrasound guidance, a 20 gauge angiocatheter was inserted through the anesthetized area into the radial artery and advanced until a flash of blood was observed in the catheter. The catheter endcap was removed, the catheter’s internal stylet withdrawn, and the catheter secured in place with gauze, tape, and a sterile transparent film dressing. One end of a 6-inch extension tubing was attached to the hub of the arterial catheter, with a three-way stopcock attached to the other end. A 500 mL warmed bag of normal saline was placed into a pressure cuff, and the output of the bag attached to the three-way stopcock via a 60-inch intravenous tubing gravity set.

#### Baseline Blood Sample Collection for Serum Analyte Analyses

Following placement of the arterial catheter, a single 8 mL arterial blood sample was obtained via the three-way stopcock, using a 10 mL syringe. Blood was then injected into an 8.5 mL BD Vacutainer^®^ tiger top serum separator tube (Benton, Dickinson & Company, Franklin Lakes, NJ, USA) and the tube was centrifuged at 3000 rpm for 8–10 min. The separated serum was pipetted into 1.5 mL Eppendorf tubes, which were immediately placed into dry ice for transport to −80 °C storage.

### 4.3. Hypobaric Chamber

Following placement of the indwelling arterial catheter, the study participant entered the human hypobaric chamber (Silvan Industries, Marinette, WI, USA). The chamber was 16 ft. long by 7 ft. wide by 7.5 ft. tall, and able to accommodate 4–6 persons. Although structurally capable of depressurizing to an atmospheric pressure of 8.17 mmHg, equivalent to an altitude of 30,480 m (100,000 feet), it was programmed to never exceed a barometric pressure below 226 mmHg, which is equivalent to an altitude of 9144 m (30,000 feet).

### 4.4. Functional Near Infrared Spectroscopy (fNIRS)

Once seated inside the chamber, an fNIRS microfiber headcap was placed securely across the participant’s scalp and connected to the NIRScout system (NIRx Medical Technologies, LLC, Glen Head, NY, USA). The fNIRS system was used to detect, measure, and quantify changes in the brain’s functional response to stimuli by measuring changes in oxygenated and deoxygenated hemoglobin in cortical areas [39]. Based upon the brain’s functional response or lack of response to stimuli, neuronal activity within discrete regions can increase, decrease, or remain the same. The subsequent level of activity modifies the volume of blood delivered to those brain regions, leading to a brief change in the concentration of oxygenated and deoxygenated hemoglobin [40]. Patterns of prefrontal cortical brain activation patterns were measured with fNIRS throughout pegboard task-on/task-off sequences.

The fNIRS headcap was equipped with 32 optodes consisting of 16 light emitting diodes (sources) and 16 receivers (detectors). Optode locations concurred with the bilateral prefrontal cortices. A dark cloth hood was then placed over the headcap to minimize ambient light intrusion to the optodes while also reducing risk of optode movement. Signal quality of optodes was confirmed using internal calibration sequences of the system’s software (nirsLAB 2017.6, NIRx Medical Technologies, LLC, Glen Head, NY, USA). If poor signal quality was detected, the hood was removed and the offending optode adjusted. Signal quality was again assessed, and those steps repeated until desired results were attained. One final measurement of signal quality was conducted after placement of the oxygen delivery mask on the participant, as described below. Continuous fNIRS data collection began at the onset of the study protocol and was recorded throughout the course of the study, which was approximately 120 mins’ duration.

#### fNIRS Analysis

Acquisition data files were downloaded from the fNIRS system computer and imported into Homer3 open-source software in MATLAB R2022a (The Mathworks, Inc., Natick, MA, USA) for analysis [41]. Preprocessing of data included selecting, from the entire data stream, only those sections occurring within the pegboard “task-on” and corresponding “task-off” periods with 2-s of data on either side. This condensed the data file into 128 discrete conditions of 20 s duration per condition of task-on or task-off. The condensed dataset was uploaded to the Homer3 GUI where a software pipeline converted raw data to “beta values” for each of the 128, 20-s discrete data conditions. Each data channel was assessed for signal strength and flagged for additional review if the signal strength or the moving average of standard deviations exceeded pre-determined threshold levels. In those cases, we employed recursive principle component analyses as defined by Yücel [42]. In addition, each channel was assessed for motion artifact. If the artifact involved multiple channels and exceeded pre-established thresholds, the entire 20-s data condition was rejected and excluded from further analyses. A high-pass and low-pass band filter was applied over the remaining data set. The optical density signals were then converted to oxyhemoglobin and deoxyhemoglobin concentrations followed by block averaging of each of the 128, 20-s data conditions. General Linear Model (GLM) analysis was then used to estimate the hemodynamic response function (HRF) [43]. Values for both oxyhemoglobin and deoxyhemoglobin were derived and exported to Statistical Analysis Software (SAS) version 9.4 (SAS Institute, Inc., Cary, NC, USA) for further statistical analysis.

Upon completion of signal processing, we then determined patterns of prefrontal cortical brain activity during pegboard testing, first immediately upon cessation of non-steady-state F_i_O_2_, and then following 120 s of exposure to steady-state F_i_O_2_. This was accomplished by comparing levels of prefrontal cortical activity during the first 20-s pegboard “task-on” time period with the following 20-s “task-off” time period. We defined the difference between levels of prefrontal cortical activity between those two conditions as “brain activation.” We then compared levels of prefrontal cortical activation between pegboard Task #1, which occurred immediately after cessation of non-steady-state F_i_O_2_, with pegboard Task #3 which occurred following 120 s of steady-state F_i_O_2_ exposure.

### 4.5. Delivery of Non-Steady-State F_i_O_2_

Following placement of the fNIRS headcap and system calibration, a gel foam-sealed facemask (Hans Rudolph Inc., Shawnee, KS, USA) was strapped securely over the participant’s nose and mouth. A 2-way non-rebreather T-shape valve (Hans Rudolph, Series 2700) was inserted into the single large bore opening on the front of the mask. The left side of the connector was attached to an inspiratory circuit while the right side of the connector was attached to the expiratory circuit. A one-way valve within the inspiratory circuit adjacent to the facemask, coupled with constant flow of gas through both the inspiratory and expiratory circuits, insured that the participant did not rebreathe previously exhaled gas.

Delivery of precise oxygen concentrations was achieved by attaching the inspiratory side of the circuit to a PM5200M Air/Oxygen Blender (Precision Medical Inc., Northampton, PA, USA). The blender was connected to compressed air (21% oxygen and 79% nitrogen) and oxygen (100%) supply lines and used to titrate specific oxygen concentrations delivered to the study participant. A BIOPAC MP160 data acquisition and analysis system (BIOPAC Systems, Inc. Goleta, CA, USA) with the BIOPAC oxygen (O_2_) 100C Module and a carbon dioxide (CO_2_) 100C Module was attached to the participant’s face mask to provide a continuous measure of actual mask levels of F_i_O_2_, CO_2_, and respiratory rate. An additional O_2_ 100C Module was connected to the analyzer to monitor ambient O_2_ levels within the hypobaric chamber.

As illustrated in Figure 1, the participant was exposed to steady-state 21% F_i_O_2_ for three minutes duration. A baseline ABG sample was then obtained from the indwelling arterial catheter using a syringe treated with dry lithium heparin. That initial ABG sample, as well as all subsequent ABG samples, were immediately passed from inside the hypobaric chamber to the outside via a small, sealed steel tube passage. Once received by a research team member outside the hypobaric chamber, the blood sample was immediately injected (within 1–2 min) into an iSTAT blood analyzer (Abbott Point of Care, Princeton, NJ, USA) equipped with a CG8+ cartridge for analysis of sodium (Na^+^), potassium (K^+^), ionized calcium (iCa^+2^), Glucose, Hematocrit, Hemoglobin, pH, partial pressure of carbon dioxide (PaCO_2_), partial pressure of oxygen (P_a_O_2_), total carbon dioxide content (TCO_2_), bicarbonate (HCO_3_^−^), base excess (BE), and oxygen saturation (SaO_2_).

### 4.6. Non-Steady-State/Steady-State F_i_O_2_ Exposures and Arterial Blood Gas Sampling

Following collection of the initial arterial blood sample, the participant executed the pegboard task for 20 s, and then rested quietly for 20 s before completing two additional pegboard task-on/task-off sequences. Following the third task, the first sequence of non-steady-state F_i_O_2_ exposure was initiated by a rapid increase from 21% to 50%, where it was maintained for 60 s, followed by a return to 21% F_i_O_2_ that was maintained for 60 s. That non-steady-state cycle was repeated a second and third time. Following the third non-steady-state cycle, the F_i_O_2_ was reduced to 35% (the midpoint between 21% and 50%) and participants completed the three task-on and task-off grooved pegboard sequences. Figure 1 illustrates that following the third oscillation of each F_i_O_2_, a 0.5 mL arterial blood sample was obtained from the arterial catheter line, followed by a 0.5 mL flush of normal saline. The F_i_O_2_ was then maintained at a steady-state level for 120 s, during which time the participant again completed the three task-on and task-off grooved pegboard sequences. At the completion of the pegboard tasks, a second (companion) 0.5 mL arterial blood sample was obtained, followed by a 0.5 mL heparinized saline (100 u/mL) flush to maintain catheter line patency. This pattern of non-steady-state F_i_O_2_ oscillations, ABG sampling, and pegboard task-on/task-off sequences was repeated at F_i_O_2_ levels of 50% ± 15%, 65% ± 15%, and 80% ± 15%, as illustrated in Figure 1. Following the final non-steady-state F_i_O_2_ exposure, the F_i_O_2_ was increased to 100% and maintained for 120 s, at which point participants completed a final series of three pegboard task-on/task-off sequences, followed by collection of a single 0.5 mL arterial blood sample.

After completing the initial 33 min exposure protocol conducted at 749 mmHg (Trial 1), the hypobaric chamber was depressurized to a barometric pressure of 565 mmHg, paralleling an altitude of 8000 ft. The same protocol of exposure to F_i_O_2_ oscillations, arterial blood sampling, and pegboard task-on/task-off sequences was again delivered over the next 33 min (Trial 2). The hypobaric chamber was then depressurized to a barometric pressure of 494 mmHg, paralleling an altitude of 15,000 ft and same exposure protocol repeated for the third and final time (Trial 3).

Following completion of Trial 3, the chamber was slowly repressurized over a 10–15 min period. Upon reaching 749 mmHg, the participant was escorted from the chamber, seated in a chair, and the arterial line removed. The participant was monitored for 30–60 min to insure absence of any potential adverse effects resulting from hypobaric conditions or arterial line placement, and then released.

### 4.7. Post-Exposure Study Visit

Within 48 h following the experimental exposure, a member of the investigative team met with the participant to assess the site of the arterial catheterization and to obtain a post-exposure blood sample for proinflammatory serum analyte analysis. At that time, a tourniquet was applied approximately 2–3 inches above the right or left antecubital vein. A 23-gauge butterfly needle with tubing was used to puncture the vein. An 8.5 mL BD Vacutainer^®^ serum separation tube was attached to the butterfly tubing, followed by collection of ~8 mL of venous blood. The venous sample was processed in the same manner as the prior arterial sample, with serum stored at −80 °C until analyses.

### 4.8. Serum Analyte Analyses

Blood serum samples were collected prior to exposure to non-steady-state F_i_O_2_ and hypobaria and again 48 h after experimental exposure. These serum samples were analyzed using multi-array technology (QuickPlex 120; MesoScale Diagnostics, Rockville, MD, USA) coupled with the V-PLEX neuroinflammation panel (Catalog No. K15210D). To establish measurement reproducibility, each sample was divided between two adjacent wells of each 96-well plate. The neuroinflammation panel included the proinflammatory plate measuring interleukin 1 beta (IL-1β), interleukin 2 (IL-2), interleukin 4 (IL-4), interleukin 6 (IL-6), interleukin 8 (IL-8), interleukin 10 (IL-10), interleukin 13 (IL-13), tumor necrosis factor alpha (TNF-α), and interferon-gamma (IFN-γ). The chemokine plate measured monocyte chemoattractant protein-1 (MCP-1), monocyte chemoattractant protein-4 (MCP-4), eotaxin, macrophage inflammatory protein-1 alpha (MIP-1α), eotaxin-3, thymus activation regulated chemokine (TARC), macrophage inflammatory protein-1 beta (MIP-1β), macrophage-derived chemokine (MDC), and interferon gamma inducible protein-10 (IP-10). The cytokine plate measured interleukin 1 alpha (IL-1α), interleukin 5 (IL-5), interleukin 7 (IL-7), interleukin 12 (IL-12), interleukin 15 (IL-15), interleukin 16 (IL-16), interleukin 17A (IL-17A), tumor necrosis factor beta (TNF-β), and vascular endothelial growth factor A (VEGF-A). The angiogenesis plate measured basic fibroblastic growth factor (bFGF), vascular endothelial growth factor C (VEGF-C), vascular endothelial growth factor D (VEGF-D), vascular endothelial growth factor receptor 1 (Flt-1), placental growth factor (PlGF), and tyrosine kinase 2 (Tie-2). The vascular injury plate measured serum amyloid A (SAA), C-reactive protein (CRP), intercellular adhesion molecule-1 (ICAM-1), and vascular cell adhesion molecule-1 (VCAM-1).

### 4.9. Statistical Approach

Comparisons of participants’ race, age, sex, and BMI were tested using Fisher’s Exact test and Wilcoxon rank sum test. fNIRS-derived measures of cortical activation levels within each brain region during pegboard Task #1 and pegboard Task #3 were explored using paired *t*-tests or Wilcoxon signed rank tests. ABG values obtained following exposures to non-steady-state F_i_O_2_ and again after steady-state F_i_O_2_ was sustained for 120 s were summarized as mean, standard deviation (SD), and range (minimum, maximum) and then compared using paired *t*-tests or Wilcoxon signed rank tests. Serum analytes measured at baseline and again at 48 h following the experimental exposures to hypobaria and to non-steady-state and steady-state FiO2 were evaluated using the same approach. The false discovery rate (FDR) method was applied to control Type I error from multiple comparison testing. All tests were two-sided and adjusted *p*-values from the FDR less than 0.05 were considered to be statistically significant. Data were analyzed using SAS version 9.4 (SAS Institute, Inc., Cary, NC, USA) and Statistical Package for Social Sciences (SPSS) version 28 (IBM Corp; Armonk, NY, USA).

## Figures and Tables

**Figure 1 ijms-25-03279-f001:**
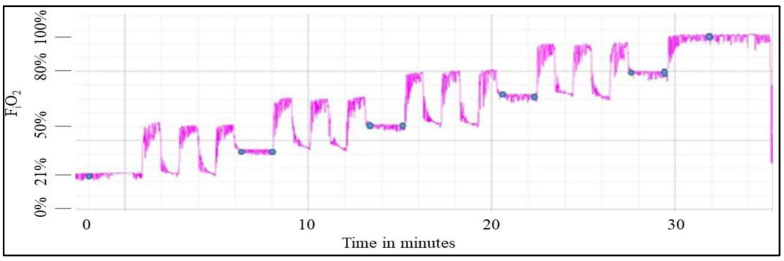
Output from mask oxygen sensor.

**Figure 2 ijms-25-03279-f002:**
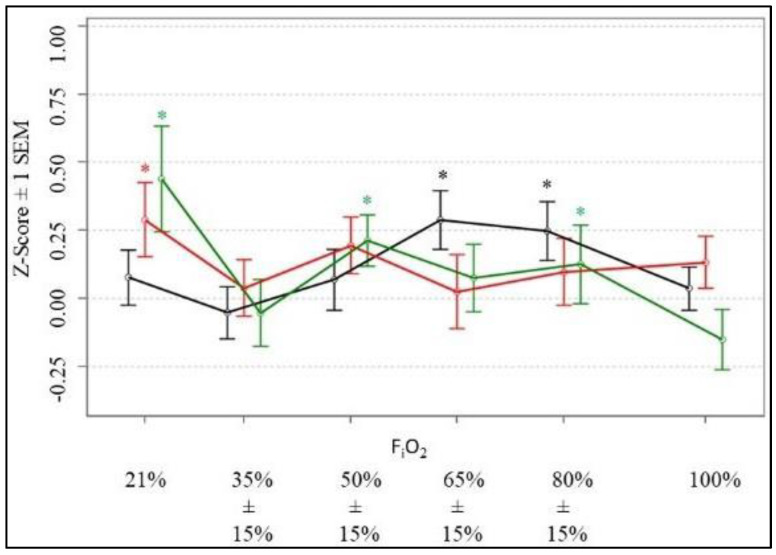
Prefrontal cortical activation. This figure illustrates the z-score ± 1 SEM of prefrontal cortical activation levels as measured by fNIRS between Task #1 and Task #3 at each barometric pressure. The black line represents data generated at 749 mmHg, red is 565 mmHg and green is 494 mmHg. The values for z-scores are staggered on the *x*-axis only for the purpose of visualization. * Represents significant differences between Task #1 and Task #3 as reported in Table 3.

**Figure 3 ijms-25-03279-f003:**
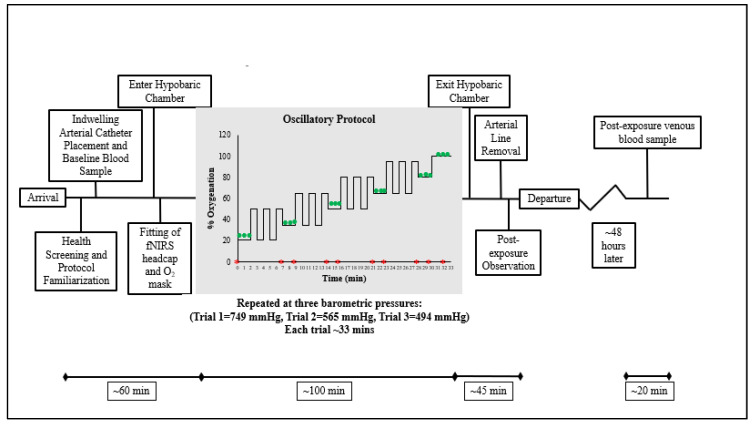
Study data collection overview. This figure illustrates the study protocol at each barometric pressure. The portion of the study in which the participant was exposed to steady-state and non-steady-state inspired oxygen is within the gray shaded box. Each exposure to non-steady-state oxygen appears as a series of cyclic square waves. Green circles represent the time points of each pegboard test. Red asterisks represent the time points of each arterial blood sample.

**Table 1 ijms-25-03279-t001:** Study participant demographics.

Sex	Race	Age (Years) M ± SD (Range)	BMI (kg/m^2^) M ± SD (Range)
**Males** n = 17	Asian = 2 Black = 5 White = 10	31.12 ± 11.03 (21–57)	28.80 ± 5.52 (22.12–45.84)
**Females** n = 7	Asian = 2 Black = 0 White = 5	29.57 ± 6.29 (24–42)	28.89 ± 5.77 (21.07–39.30)
*p*-value	0.2020 ^a^	0.8731 ^b^	0.9241 ^b^

No age or BMI differences were found between males and females. ^a^ *p*-value from Fisher’s exact test. ^b^ *p*-values from Wilcoxon rank sum test.

**Table 2 ijms-25-03279-t002:** Arterial blood P_a_O_2_ comparisons.

	F_i_O_2_ Range during Non-Steady-State/Steady-State Hyperoxia Exposure	N ^†^	P_a_O_2_ Measured upon Reaching Predetermined F_i_O_2_ Level Following Non-Steady-State/Steady-State Hyperoxia Exposure (Mean ± SD) Range	P_a_O_2_ Measured after Maintaining Predetermined F_i_O_2_ Level for 120 s (Mean ± SD) Range	*p*-Value	Adj. *p*-Value
**749 mmHg**	21% steady state	24		95.63 ± 13.21 (68–122)	n/a	
(% Change)	35% ± 15%	24	216.67 ± 29.53	178.25 ± 26.73	<0.0001	<0.0001
17.73%			(115–264)	(90–210)		
13.75%	50% ± 15%	24	296.92 ± 37.53	255.50 ± 34.59	<0.0001	<0.0001
			(166–356)	(121–288)		
9.73%	65% ± 15%	24	382.50 ± 42.99	345.29 ± 44.56	<0.0001	<0.0001
			(227–431)	(171–396)		
10.01%	80% ± 15%	24	470.96 ± 55.59	423.83 ± 49.81	<0.0001 *	<0.0001
			(318–548)	(225–471)		
	100% steady state	24		541.83 ± 47.62 (385–613)	n/a	
**565 mmHg**	21% steady state	24		66.71 ± 6.10 (51–83)	n/a	
21.02%	35% ± 15%	24	154.21 ± 19.72	121.79 ± 17.99	<0.0001	<0.0001
			(105–195)	(78–163)		
18.79%	50% ± 15%	24	218.25 ± 25.10	177.25 ± 27.34	<0.0001	<0.0001
			(134–264)	(77–208)		
11.68%	65% ± 15%	23	274.35 ± 22.37	242.30 ± 27.11	<0.0001	<0.0001
			(218–301)	(161–273)		
9.97%	80% ± 15%	23	336.52 ± 41.60	302.96 ± 40.47	<0.0001	<0.0001
			(174–391)	(141–342)		
	100% steady state	23		379.29 ± 46.56 (189–420)	n/a	
**494 mmHg**	21% steady state	23		47.17 ± 5.51 (36–58)	n/a	
27.26%	35% ± 15%	23	107.65 ± 16.38	78.30 ± 10.61	<0.0001	<0.0001
			(77–136)	(53–93)		
18.09%	50% ± 15%	23	153.83 ± 20.75	126.00 ± 17.22	<0.0001	<0.0001
			(93–186)	(79–152)		
13.64%	65% ± 15%	23	199.26 ± 25.32	172.09 ± 23.19	<0.0001	<0.0001
			(111–229)	(94–197)		
11.21%	80% ± 15%	23	242.39 ± 21.26	215.22± 28.16	<0.0001	<0.0001
			(184–267)	(112–237)		
	100% steady state	23		273.22 ± 29.27 (184–306)	n/a	

Comparisons of arterial blood P_a_O_2_ sampled immediately following 5 min of exposure to oscillatory F_i_O_2_ were made with those sampled immediately following 120 s of exposure to steady-state F_i_O_2_. Comparisons were performed using paired *t*-tests unless otherwise indicated. * *p*-values from Wilcoxon signed rank test; adj. *p*-values from false discovery rate (FDR). ^†^ Sample size for ABG measurements decreased as a result of one participant whose arterial line catheter became non-functional during the latter part of the experimental protocol.

**Table 3 ijms-25-03279-t003:** Prefrontal cortical activation levels.

	F_i_O_2_ Range during Non-Steady-State/Steady-State Hyperoxia Exposure	N ^†^	Activation Level during Pegboard Task #1 (Mean ± SD) Range	Activation Level during Pegboard Task #3 (Mean ± SD) Range	*p*-Value	Adj. *p*-Value
**749 mmHg**	21% steady state	23	0.992 ± 0.918	1.02 ±1.051	0.4625	0.6508
			(−0.873–2.525)	(−1.009–2.676)		
	35% ± 15%	24	0.779 ± 1.029	0.726 ± 0.955	0.5872	0.6508
			(−0.692–2.81)	(−1.153–2.576)		
	50% ± 15%	24	0.662 ± 1.061	0.731 ± 0.822	0.5438	0.6508
			(−0.999–2.954)	(−0.562–2.317)		
	65% ± 15%	24	0.632 ± 0.932	0.92 ± 0.871	0.0146	0.0876
			(−1.36–2.692)	(−0.356–2.892)		
	80% ± 15%	24	0.609 ± 0.815	0.856 ± 0.885	0.0321	0.0963
			(−0.778–2.176)	(−1.071–2.649)		
	100% steady state	23	0.431 ± 0.969	0.467 ± 0.931	0.6508	0.6508
			(−1.293–2.782)	(−1.243–2.988)		
**565 mmHg**	21% steady state	22	0.476 ± 0.987	0.741 ± 1.081	0.0466	0.2190
			(−1.154–2.601)	(−0.723–3.614)		
	35% ± 15%	22	0.756 ± 0.673	0.794 ± 0.845	0.7181	0.8554
			(−0.17–2.324)	(−0.673–2.449)		
	50% ± 15%	24	0.501 ± 1.048	0.695 ± 0.965	0.0730	0.2190
			(−1.548–2.951)	(−1.375–2.322)		
	65% ± 15%	22	0.662 ± 1.056	0.687 ± 0.945	0.8554	0.8554
			(−1.177–4.221)	(−0.819–3.677)		
	80% ± 15%	23	0.555 ± 1.055	0.592 ± 1.031	0.4351	0.6527
			(−1.817–2.549)	(−1.769–2.381)		
	100% steady state	23	0.791 ± 0.885	0.887 ± 0.916	0.1820	0.3640
			(−1.223–2.455)	(−1.675–2.36)		
**494 mmHg**	21% steady state	22	0.434 ± 1.006	0.874 ± 0.764	0.0340	0.0680
			(−1.577–2.639)	(−0.631–2.035)		
	35% ± 15%	23	0.997 ± 1.076	0.943 ± 1.061	0.6663	0.6663
			(−1.041–3.164)	(−0.749–3.776)		
	50% ± 15%	21	0.666 ± 0.791	0.879 ± 1.071	0.0335	0.0680
			(−0.526–2.466)	(−0.674–3.611)		
	65% ± 15%	22	0.798 ± 0.948	0.872 ± 0.949	0.5564	0.6663
			(−0.558–2.968)	(−0.942–2.829)		
	80% ± 15%	22	0.605 ± 0.949	0.726 ± 0.915	0.0271 *	0.0680
			(−0.715–2.583)	(−0.47–3.205)		
	100% steady state	22	0.762 ± 1.075	0.61 ± 0.992	0.1838 *	0.2757
			(−1.455–2.721)	(−1.361–2.439)		

fNIRS contrasts performed during the grooved pegboard task immediately following exposure to oscillatory F_i_O_2_ were compared with those performed during a grooved pegboard task following 120 s of exposure to steady-state F_i_O_2_. Comparisons were performed using paired *t*-tests unless otherwise indicated. ^†^ Sample size variation is due to artifact rejection which deleted some sequences of fNIRS data. * *p*-values from Wilcoxon signed rank test.

**Table 4 ijms-25-03279-t004:** Serum analytes.

Analyte	N ^†^	Baseline M ± S.D (Range)	Post-Hypobaria and Hyperoxia Exposure M ± S.D (Range)	*p*-Value ^a^	Adj. *p*-Value
**Proinflammatory Plate (pg/mL)**					
**IL-6**	23	1.367 ± 3.201	1.973 ± 3.790	0.014 *	0.0268
		(0.010–15.898)	(0.305–17.981)		
**Chemokine Plate (pg/mL)**					
**IP-10**	23	146.535 ± 76.353	245.677 ± 183.548	<0.0001 *	0.0004
		(34.809–394.060)	(91.328–770.989)		
**MCP-1**	23	84.383 ± 43.204	119.536 ± 60.249	0.023	0.0268
		(15.264–152.951)	(46.300–303.390)		
**MDC**	23	*495.726* ± *204.249*	592.515 ± 207.538	0.004	0.0140
		(160.135–1066.029)	(324.178–1015.837)		
**Cytokine (pg/mL)**					
**IL-15**	23	1.464 ± 0.613	1.671 ± 0.558	0.023	0.0268
		(0.678–2.832)	(0.817–2.628)		
**Angiogenesis (pg/mL)**					
**VEGF-D**	23	1423.960 ± 343.450	1529.171 ± 417.718	0.023	0.0268
		(853.415–2225.042)	(824.415–2636.861)		

Six serum analyte levels changed significantly from baseline when compared to serum analyte levels measured following exposure to hypobaria and hyperoxia. ^†^ N = 23, due to one missing post-exposure venous blood sample; * *p*-values from Wilcoxon signed rank test; ^a^ *p*-values from paired *t*-tests; adj. *p*-values from false discovery rate (FDR).

## Data Availability

Data are subject to third party restrictions. The data are not openly available due to privacy or ethical restrictions. Data supporting the findings of this study may be available from the United States Air Force upon reasonable request submitted to the corresponding author.

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
