# Peer review of "Exposure to Non-Steady-State Oxygen Is Reflected in Changes to Arterial Blood Gas Values, Prefrontal Cortical Activity, and Systemic Cytokine Levels"

_ijms, 2024, doi:10.3390/ijms25063279_

Round 1
Reviewer 1 Report
Comments and Suggestions for Authors
minor editing of English language required
Author Response
Thank you for taking the time to review our manuscript and to make comments.
Your two queries may be better answered here rather than in the manuscript.
The first question of "Please explain why the number of participants of male are higher?"
The answer is that the twice as many males volunteered for the study than females. Our hope was to achieve equal numbers of males and females. However, we found that far fewer females expressed interest in participation.
The second question was "And please explain about age are chosen."
The answer is that candidate study participants were required to be between the ages of 18-61 years. Any person younger than 18 is considered a "child" by OSHA guidelines. Any person > age 61 was considered, by our IRB, to be "too old" to participate in a hypobaric research study.
2. Please check that all reference’s format.
Thank you for the suggestion. We have reviewed the references ad they appear to be in order. Many citations included six or more authors, so by convention, only the first author's name appears and is followed by et al.
We have reformatted the references to include the names of all author's as the Journal Guidelines permit up to 10 names.
Thank you, once again, for your time reviewing and commenting on our project.
Reviewer 2 Report
Comments and Suggestions for Authors
Elegant study.
Line 139: “…(Table2)” is not necessary.
Lines 371 and 423: use ABG instead of “…arterial blood gas…”
Author Response
Thank you for taking the time to review our manuscript and to make comments.
We appreciate your suggestion to change "Arterial Blood Gas" to "ABG" in several locations. We have done this.
We have also deleted the words "Table 2" as you suggested.
Thank you, once again, for your time reviewing and commenting on our project.
Reviewer 3 Report
Comments and Suggestions for Authors
Damato and co-authors conducted a study on how non steady-state inspiration of oxygen impacts measured arterial oxygen levels, prefrontal cortical activation (via fNIRS + cognitive tests), and levels of select serum analytes for inflammation and angiogenesis during normobaric (749 mmHg) and hypobaric (565 & 494 mmHg) environmental pressures. The authors found that cessation of non-steady state levels of inspired oxygen >50% resulted in higher arterial oxygen levels relative to cessation of steady-state exposures. This general effect persisted across all inspired oxygen levels at each barometric pressure. Further, prefrontal cortical activation during cognitive tests was lower following non steady-state oxygen exposure (>50% & <100%) during normobaric and hypobaric (494 mmHg) conditions. Finally, serum analyte levels of select markers (e.g., IL-6, IP-10, VEGF-D) increased after 48 hours of hypobaria + hyperoxia as well. In light of the clear presentation of the study and limitations managed for discussion, I do not have recommendations for further improvement of the manuscript.
Author Response
Thank you for taking the time to review our manuscript and to make comments.
We agree that the research design is very intricate as it included multiple simultaneous experimental manipulations with corollary outcome measures. With so many inherent complexities, we found it very challenging to articulate the design simple straightforward manner.
We could try to rewrite the entre Methods section, but that will require that we cancel the submission process. The project was funded by the USAF, and they have reviewed and approved the current manuscript. However, any substantive changes require that we rescind the submission, make necessary changes, and then resubmit the manuscript to the USAF for their approval. Once the the USAF has reviewed and approved, we can resubmit the revised manuscript for re-review.
If you feel that process is necessary, we are happy to do so if you feel the manuscript is unacceptable in it's current condition.
Thank you, once again, for your time reviewing and commenting on our project.